# Electrical spin injection and detection in molybdenum disulfide multilayer channel

Shiheng Liang[1], Huaiwen Yang[1], Pierre Renucci[2], Bingshan Tao[1], Piotr Laczkowski[3], Stefan Mc-Murtry[1], Gang Wang[2], Xavier Marie[2], Jean-Marie George[3], Sébastien Petit-Watelot[1], Abdelhak Djeffal[1], Stéphane Mangin[1], Henri Jaffrès[3] & Yuan Lu[1]

Molybdenum disulfide has recently emerged as a promising two-dimensional semiconducting material for nano-electronic, opto-electronic and spintronic applications. However, the demonstration of an electron spin transport through a semiconducting $MoS_2$ channel remains challenging. Here we show the evidence of the electrical spin injection and detection in the conduction band of a multilayer $MoS_2$ semiconducting channel using a two-terminal spin-valve configuration geometry. A magnetoresistance around 1% has been observed through a 450 nm long, 6 monolayer thick $MoS_2$ channel with a Co/MgO tunnelling spin injector and detector. It is found that keeping a good balance between the interface resistance and channel resistance is mandatory for the observation of the two-terminal magnetoresistance. Moreover, the electron spin-relaxation is found to be greatly suppressed in the multilayer $MoS_2$ channel with an in-plane spin polarization. The long spin diffusion length (approximately ∼235 nm) could open a new avenue for spintronic applications using multilayer transition metal dichalcogenides.

[1] Institut Jean Lamour, UMR 7198, CNRS-Université de Lorraine, BP 239, 54506 Vandœuvre, France. [2] Université de Toulouse, INSA-CNRS-UPS, LPCNO, 135 avenue de Rangueil, 31077 Toulouse, France. [3] Unité Mixte de Physique, CNRS, Thales, Univ. Paris-Sud, Université Paris-Saclay, 91767 Palaiseau, France. Correspondence and requests for materials should be addressed to Y.L. (email: yuan.lu@univ-lorraine.fr) or to H.J. (email: henri-yves.jaffres@cnrs-thales.fr).

Transition metal dichalcogenides (TMDs) have emerged as a promising 2D crystal family, demonstrating solutions for several novel nano-electronic and opto-electronic applications[1–7]. In contrast to graphene and boron nitride, which are respectively a metal and a wide-gap semiconductor, TMDs family displays a large variety of electronic properties ranging from semiconductivity to superconductivity[8]. As a representative of TMDs, molybdenum disulfide ($MoS_2$) has a tunable bandgap that changes from an indirect gap of 1.2 eV in the bulk to a direct gap of 1.8 eV for one monolayer (ML)[1]. The ML $MoS_2$ is characterized by a large spin-orbit splitting of ~0.15 eV in the valence band[3,4] together with a small value of ~3 meV for the conduction band[9]. The lack of inversion symmetry combined with the spin-orbit interaction leads to a unique coupling of the spin and valley degrees of freedom, yielding robust spin and valley polarizations[4–7].

To realize electron spin transport in the vicinity of the conduction band of $MoS_2$ channel, one of the prerequisites is the investigation of the electron spin-relaxation mechanism within the host material. For ML $MoS_2$, both the intrinsic spin splitting of the valence band and the Rashba-like spin-orbit coupling (SOC) due to the breaking of the inversion symmetry along the growth direction favour the spin transport through $MoS_2$ with an out-of-plane spin polarization[10]. This is because that the SOC creates an equivalent perpendicular $k$-dependent magnetic field due to the Dresselhaus effective interactions and it is associated to the D'yakonov-Perel (DP) spin relaxation mechanism[11]. If electrons with in-plane spin polarization are injected into ML $MoS_2$, the effective magnetic field can induce an efficient in-plane spin precession along the field[12] as well as the spin-dephasing. Consequently, this yields a predicted short spin lifetime (10–200 ps)[13] together with a small spin diffusion length (~20 nm)[14]. Recently, a hole spin injection into ML TMDs has

been demonstrated either with perpendicular magnetized GaMnAs injector[15] or NiFe injector at large perpendicular magnetic field[16] by electrical injection and optical detection method. This particularly emphasizes on the difficult issue to electrically inject and detect electron with in-plane spin polarization in a lateral ML $MoS_2$ device. To avoid the DP spin relaxation, one solution is to recover the inversion symmetry with thicker multilayer $MoS_2$. The recent measurement of the second-harmonic generation (SHG) efficiency as a function of the number of MLs is a good probe of the TMDs material symmetry[17]. For one ML, a strong SHG is detected due to the lack of inversion symmetry[18]. However, in the case of bilayers and 4ML $MoS_2$, the magnitude of SHG signal decreases by three orders of magnitude due to the recovery of inversion symmetry. Longer spin relaxation time can be expected for such structures. Thus, we consider only multilayer $MoS_2$ to demonstrate the in-plane electrical spin injection and detection.

Here, we provide a clear demonstration of a robust spin-valve magnetoresistance (MR) (1.1%) through the conduction band of 6ML thick $MoS_2$ channel with ferromagnetic (FM) Co/MgO tunnel injector at low temperature. This occurs in the optimal experimental situation of impedance matching between the interface resistance and the channel resistance. The clear spin-injection signals in $MoS_2$ demonstrate a spin-transport in $MoS_2$ with a relative long spin-diffusion length larger than 200 nm. This could open future avenues to use multilayer TMDs as an in-plane spin transport template and the electron spin can be manipulated by SOC for a well-defined $MoS_2$ thickness controlled by plasma etching technique[19].

## Results

**$MoS_2$ contact and channel resistance.** A key issue for electrical spin injection is the conductivity mismatch between the

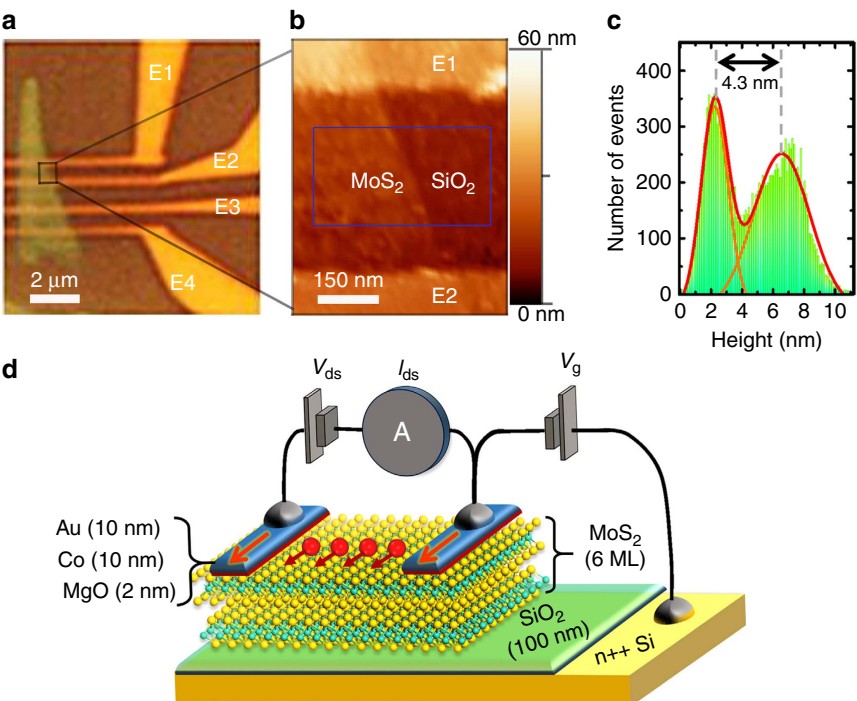

**Figure 1 | Multilayer $MoS_2$-based lateral spin-valve device. (a)** Optical image of the device with the multilayer $MoS_2$ flake on 100 nm $SiO_2$/Si(n + +) substrate, the E1, E2, E3 and E4 indicate the four Au/Co/MgO electrodes. **(b,c)** AFM measurement (in tapping mode) focused on the $MoS_2$ channel between E1 and E2 electrodes. The thickness of $MoS_2$ is determined by the Gaussian distribution of pixel height in the square region in **b**. **(d)** Schematics of the lateral spin-valve device. The multilayer $MoS_2$ serves as a spin transport channel, and two Au/Co/MgO electrodes are used to inject spin ($V_{ds}$) and measure the current ($I_{ds}$). A back-gate voltage ($V_g$) between the substrate and one top contact is used to modulate the carrier density in the $MoS_2$ channel.

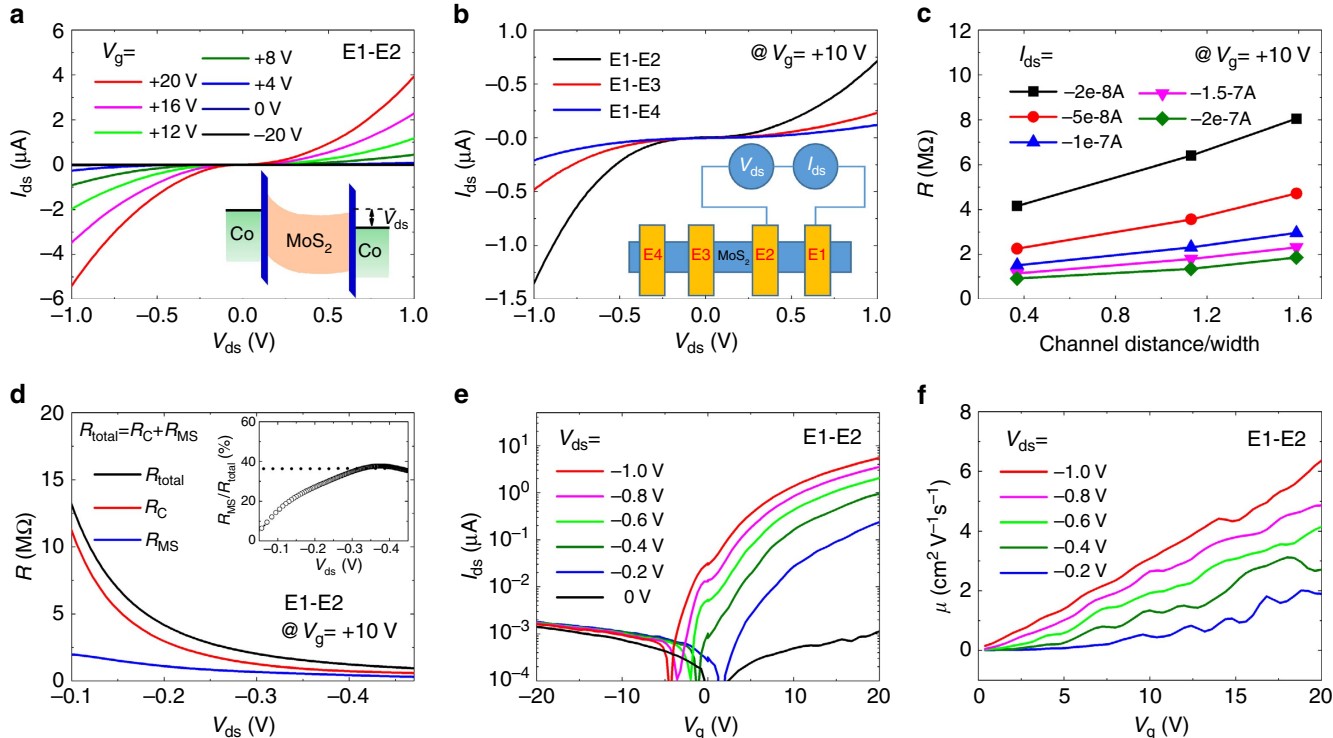

**Figure 2 | Transport characterization of MoS₂ field-effect transistor.** (**a**) Current ($I_{ds}$)–voltage ($V_{ds}$) characteristics between the E1 and E2 electrodes, measured at 12 K with applying different back-gate voltages $V_g$. Inset: band diagram of the back-to-back diode structure of the MoS₂ device with a two-terminal bias $V_{ds}$. (**b**) $I_{ds}$–$V_{ds}$ characteristics measured between different electrodes with a back-gate voltage $V_g = +10$ V at 12 K. Inset: schematics of connection between different electrodes. (**c**) The total resistance ($R$) between the two electrodes versus the channel distance (normalized by the width) with different $I_{ds}$. (**d**) $V_{ds}$ dependence of the total resistance (E1–E2) and the extracted contact resistance $R_C$ and the MoS₂ channel resistance $R_{MS}$ (E1–E2). Inset: The contribution of $R_{MS}$ in the total resistance as a function of $V_{ds}$. (**e**) Transfer characteristic $I_{ds}$–$V_g$ between E1 and E2 electrodes, measured at 12 K with applying different $V_{ds}$. (**f**) Extracted effective mobility $\mu$ versus $V_g$ with different $V_{ds}$.

FM injector and the semiconducting MoS₂ channel, which generally results in a vanishing MR[20,21] due to the spin-backflow processes[21]. In FM/MoS₂ contacts, a Schottky barrier (SB) height ($\Phi_b$) of 100–180 meV is generally created at the interface with an extended depletion region[22,23]. However, it has been recently demonstrated that an efficient reduction of $\Phi_b$ down to ∼10 meV at zero back-gate voltage can be achieved by inserting a 1–2 nm layer of MgO (ref. 23), Al₂O₃ (ref. 24) or TiO₂ (ref. 25) as a thin tunnel barrier between the FM and MoS₂. A careful design of the interface structure consisting of an oxide tunnel barrier injector (MgO) on top of an unavoidable Schottky contact thus appears mandatory to get efficient electrical spin injection and spin-detection[26].

In our devices, MoS₂ flakes were mechanically exfoliated onto a SiO₂/Si (n++) substrate. Four FM contacts composed of MgO (2 nm)/Co (10 nm)/Au (10 nm) were deposited on one MoS₂ flake (see details in Methods). The four electrodes have almost identical width around 300 nm with channel distances varying from 450 to 2,800 nm (Fig. 1a). The thickness of the flake is determined by atomic force microscopy characterization to be about 4.3 nm (Fig. 1b,c). Considering 0.72 nm for one ML MoS₂ (ref. 27), the thickness of the flake corresponds to 6ML MoS₂. Figure 1d shows schematics of the device. A drain-source bias ($V_{ds}$) between two top contacts was applied to inject a current $I_{ds}$. Meanwhile, a back-gate voltage ($V_g$) was applied between the substrate and one top contact to modulate the carrier density in the MoS₂ channel.

Let us first focus on the two-terminal $I_{ds}$–$V_{ds}$ characteristics at 12 K between electrodes E1 and E2 with different $V_g$ (Fig. 2a).

At $V_g = 0$ V, the current level is rather low ($I_{ds} \sim -100$ nA at $V_{ds} = -1$ V). By applying a back-gate voltage, a large increase of the current is observed with positive $V_g$, while the current density is greatly suppressed at negative $V_g$. The quasi-symmetric nonlinearity of $I_{ds}$–$V_{ds}$ is attributed to the back-to-back Schottky diode structures of the device (inset of Fig. 2a), thus indicating the important role played by the Co/MgO/MoS₂ Schottky contacts. The contact region characterized by the contact resistance ($R_C$) is constituted by the MgO tunnel barrier and the depletion zone of MoS₂ underneath. To extract the respective contribution of the contact resistance and MoS₂ channel resistance, we have acquired $I_{ds}$–$V_{ds}$ curves at $V_g = +10$ V with different channel distances (Fig. 2b). Those emphasize on a larger contribution of $R_C$ in the total resistance ($R_{total}$) with a shorter channel distance. Since the resistance of the MoS₂ channel ($R_{MS}$) is proportional to the channel distance, $R_C$ can be extracted from the intercept of linear fitting of the total resistance as a function of channel distance if one assumes a constant contact resistance (see details in Supplementary Note 1). To ensure an identical voltage drop on the contact region, we have plotted in Fig. 2c $R_{total}$ versus the channel distance (normalized by its width) at different $I_{ds}$. In Fig. 2d, the extracted value of $R_C$ rapidly drops off on increasing $|V_{ds}|$ whereas it dominates the total resistance at low $V_{ds}$. The large variation of $R_C$ is ascribed to the change of the Schottky profile versus $V_{ds}$. This is a major issue of our devices as discussed in the following. The extracted MoS₂ channel resistance ($R_{MS}$) displays a smaller variation on increasing $|V_{ds}|$. The ratio of $R_{MS}/R_{total}$ becomes saturated to be about 35% when $|V_{ds}| > 0.35$ V (inset of Fig. 2d). If one assumes that

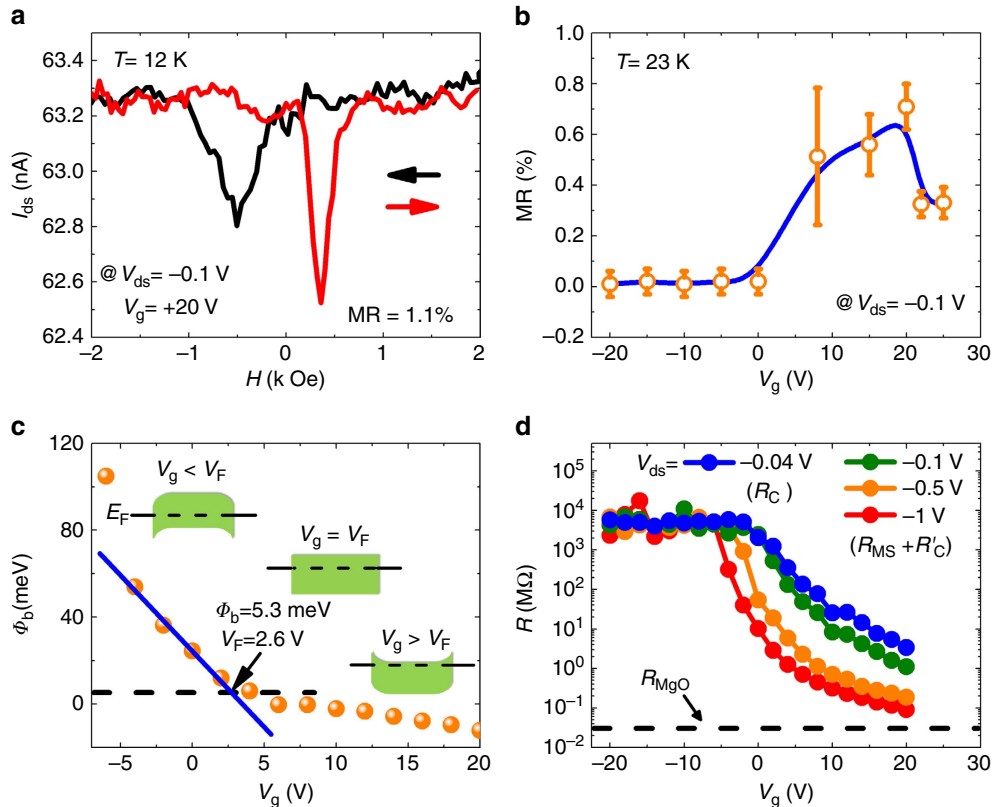

**Figure 3 | Back-gate voltage-dependent MR characterization of the device.** (**a**) Magneto-resistance response of the multilayer MoS$_2$-based lateral spin-valve device measured at 12 K with $V_g = +20$ V and $V_{ds} = -0.1$ V. (**b**) Back-gate voltage dependence of MR measured at 23 K with $V_{ds} = -0.1$ V. The error bars have been calculated by taking account of the signal noise and the contribution of leakage current. (**c**) Back-gate voltage-dependent Schottky barrier height ($\Phi_b$). The deviation from the linear response at low $V_g$ (blue solid line) defines the flat band voltage ($V_F$) and the real $\Phi_b$ of Co/MgO on MoS$_2$. Insets: schematics of MoS$_2$ band structure with different $V_g$. (**d**) Variation of the total resistance ($R$) as a function of $V_g$ with different $V_{ds}$ at 23 K. The error bars in **b** have been calculated by taking account of the signal noise and the contribution of leakage current.

this ratio is still valid at $|V_{ds}| = 1$ V, $R_{MS}$ can be estimated to be 102 kΩ for $V_g = +10$ V and 32.8 kΩ for $V_g = +20$ V (sheet resistance $R_{sq} \sim 1 \times 10^5 \Omega$). At $|V_{ds}| = 1$ V and $V_g = +20$ V, $R_C$ approaches $2R_{MgO} = 63.7$ kΩ corresponding to two intrinsic MgO barriers in series free of depletion tunnelling zone, thus giving $R_{MgO} = 31.8$ kΩ.

In Fig. 2e, $I_{ds}$ as a function of $V_g$ is plotted for different $V_{ds}$. The transistor ON/OFF ratio can be estimated from the current ratio between $V_g = \pm 20$ V, which is around $2 \times 10^3$. The lower ON/OFF ratio compared to the reported values[2] is due to the influence of leakage current on $I_{ds}$ ($\sim 1.5$ nA at $V_g = \pm 20$ V) in the OFF state because of the slight damage of contacts during wire bonding (see Supplementary Note 2). The effective field-effect mobility $\mu$ can be estimated by extracting the slope $dI_{ds}/dV_g$ from the $I_{ds} - V_g$ curves (Fig. 2e):

$$\mu = \frac{dI_{ds}}{dV_g} \frac{L}{w C_i V_{ds}} \quad (1)$$

where $L$ is the channel length, $w$ is the channel width and $C_i$ is the gate capacitance[2]. It is found that $\mu$ increases with $V_g$ as well as $V_{ds}$ (Fig. 2f). At $V_g = +20$ V and $V_{ds} = -1$ V, the mobility equals $\mu \sim 6$ cm$^2$ V$^{-1}$ s$^{-1}$ in close agreement with previously reported value (7 cm$^2$ V$^{-1}$ s$^{-1}$) at 10 K for ML MoS$_2$ on SiO$_2$/Si substrate[28]. In this low temperature range the transport in MoS$_2$ is dominated by scatterings on charged impurities[28] or hopping through localized states[29].

**Magnetoresistance measurements.** We now focus on the key results of this paper about the MR experiments on MoS$_2$. These are performed at relatively low bias ($|V_{ds}| < 0.15$ V) in order to avoid inherent hot-electrons spin depolarization mechanisms. Figure 3a displays the recorded magneto-current curve at 12 K with $V_{ds} = -0.1$ V and $V_g = +20$ V. A clear spin-valve signal is observed characterized by a larger current flowing in the parallel (P) state at high field and a smaller current in the quasi-antiparallel (AP) magnetic state at low field. The MR ratio can be calculated from $(I_P - I_{AP})/I_{AP} \times 100\%$ to be about 1.1%. This result constitutes a clear demonstration of an electron spin transport through the conduction band of MoS$_2$ in a lateral geometry. We have carefully checked the angle dependence of MR, leakage current and electrode resistance to rule out any possible spurious effects on MR such as charged impurities in MoS$_2$, spin transport in the Si substrate or anisotropic MR effect of Co electrodes (see Supplementary Note 3). In addition, micromagnetic simulations prove a comparable coercivity field as observed in our experimental configuration (see Supplementary Note 4). Reproducible results have been obtained on several samples and no spin signal can be found in the control non-magnetic samples (see Supplementary Note 3). All of these confirm that the MR results originate from spin transport from MoS$_2$.

A very interesting feature is the particular back-gate voltage dependence of MR near the optimal condition for spin-injection/detection. Figure 3b displays MR versus $V_g$ showing a characteristic maximum MR signal at a gate voltage of $V_g = +20$ V. In order to clarify this point, one should first

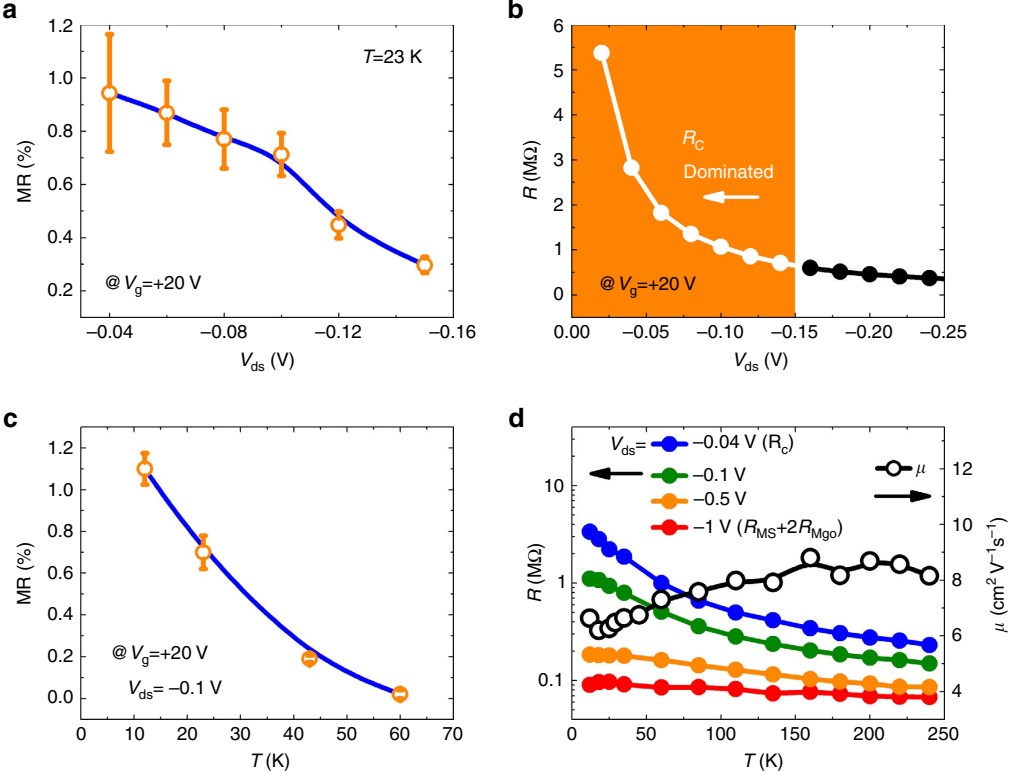

**Figure 4 | Drain-source bias and temperature-dependent MR characterization of the device.** (**a**) $V_{ds}$ dependence of MR measured at 23 K with $V_g = +20$ V. (**b**) The total resistance ($R$) of the device versus $V_{ds}$. The area with orange colour indicates the $V_{ds}$ range where the total resistance is dominated by the contact resistance. (**c**) Temperature dependence of MR measured with $V_g = +20$ V and $V_{ds} = -0.1$ V. (**d**) Temperature dependence of the total resistance ($R$) with different $V_{ds}$ and the MoS$_2$ channel mobility $\mu$ ($V_g = +20$ V, $V_{ds} = -1$ V). The error bars in **a**,**c** have been calculated by taking account of the signal noise and the contribution of leakage current.

understand the effect of $V_g$ on the transport properties. As shown in the inset of Fig. 3c, the back-gate mainly plays two roles. One is to modulate the Fermi level ($E_F$) inside the MoS$_2$ bandgap yielding a change of the carrier density in the channel[2]. The second role is to modify the SB profile and width. This scenario can be supported by the measurement of the back-gate dependence of the SB height ($\Phi_b$) of Co/MgO on MoS$_2$, as shown in Fig. 3c (see Supplementary Note 5). It is noted that all $\Phi_b$ values are extracted from the measurements between 180 K to 240 K and with $V_{ds}$ from $-0.4$ V to $-1$ V. Here, we can identify two regions for the variation of $\Phi_b$ versus $V_g$. For $V_g < +2.6$ V, that is when the depletion layer is thick, the thermionic emission dominates and this results in a large linear increase of $\Phi_b$ on the negative $V_g$. However, for $V_g > +2.6$ V, the tunnel current through the thin SB impinges on the linearity of $\Phi_b$. The real value of $\Phi_b$ for Co/MgO on MoS$_2$ is obtained at the point of the onset of the deviation ($+2.6$ V) equalling thus 5.3 meV in good agreement with the value reported for Co/Al$_2$O$_3$ (2.5 nm) on multilayer MoS$_2$ (ref. 24). This analysis would provide the SB height only when the thermal activation is dominant, that is, below $+2.6$ V. Above this value, the analysis can still be performed, but it does not lead to the interpretation of SB height, since tunnelling is dominant. In Fig. 3d, we have plotted the total resistance versus $V_g$ for different $V_{ds}$. As mentioned above, the resistance is mainly attributed to the contact resistance $R_C$ at $V_{ds} = -0.04$ V. At $V_{ds} = -1$ V, the contribution from $R_{MS}$ can reach 35% in $R_{total}$ when the depletion region is much reduced. At large negative $V_g$ when $E_F$ is far away from the bottom of the conduction band, the resistance is rather high and does not vary

with $V_{ds}$ certainly preventing any spin transport in MoS$_2$. For positive $V_g$, the channel and contact resistance both decrease rapidly with $V_g$ when $E_F$ moves close to the conduction band. From $-20$ V to $+20$ V, the MoS$_2$ channel resistance decreases much faster than the contact resistance ($5 \times 10^4$ times for $V_{ds} = -1$ V versus $1.2 \times 10^3$ times for $V_{ds} = -0.04$ V).

The bias dependence of MR measured with $V_g = +20$ V at 12 K is displayed in Fig. 4a. It is found that the MR ratio decreases with the increase of bias $|V_{ds}|$. When $|V_{ds}|$ is larger than 0.15 V, MR almost disappears. We note that the total resistance also decreases rapidly with the increase of $|V_{ds}|$ (Fig. 4b), especially in the range $|V_{ds}| < 0.15$ V where $R_C$ is considered to be dominant as mentioned above. This indicates that the observation of MR could be related to the large contact resistance introduced to overcome the impedance mismatch at Co/MgO/MoS$_2$ interface. In Fig. 4c, we display the temperature ($T$) dependence of MR acquired with $V_g = +20$ V and $V_{ds} = -0.1$ V: the MR rapidly decreases with $T$. When $T > 60$ K, MR almost disappears. This thermal behaviour could be also associated to the variation of $R_C$ versus $T$ if one assumes a constant spin polarization at Co/MgO interface[30]. In Fig. 4d, the total resistance as a function of $T$ is plotted for different $V_{ds}$ conditions. It appears obviously that the resistance at $V_{ds} = -0.04$ V ($R_C$ dominant) decreases rapidly when $T > 60$K, while the resistance with $V_{ds} = -1$ V ($R_{MS} + 2R_{MgO}$) decreases more slowly with $T$. Therefore, a strong correlation between the contact resistance and MR is also revealed from the temperature dependence of MR. In Fig. 4d, we also show the temperature variation of mobility derived at $V_g = +20$ V and

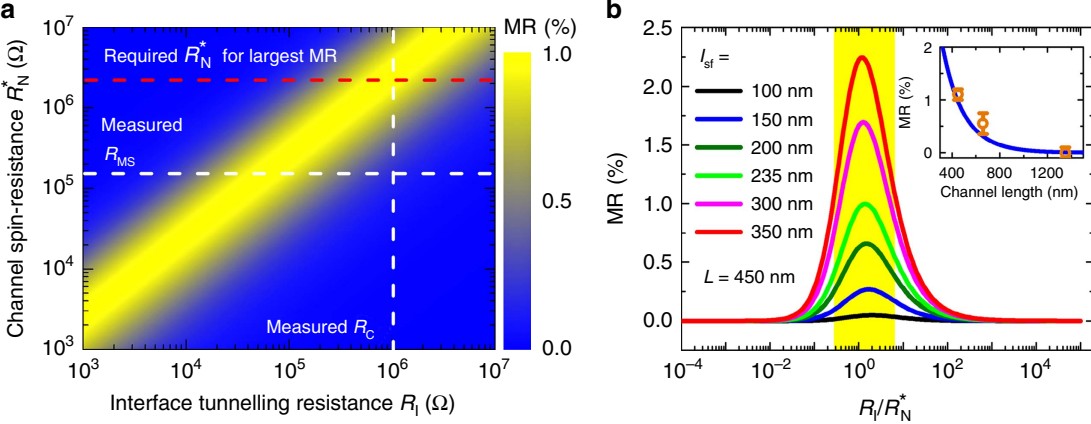

**Figure 5 | Calculation of MR with spin diffusion theory.** (**a**) Calculated MR of FM/I/MoS$_2$/I/FM structure as a function of the interface tunneling resistance ($R_I$) and the channel spin-resistance ($R_N^\star$). (**b**) Calculated MR as a function of the ratio $R_I/R_N^\star$ with different spin diffusion length ($l_{sf}$). Inset: theoretical calculated MR versus channel length (blue line) and experimentally observed MR between different electrodes (E1–E2, E2–E3 and E1–E3). The error bars have been calculated by taking account of the signal noise and the contribution of leakage current.

$V_{ds} = -1$ V. It is noted that the mobility of MoS$_2$ channel shows a slight increase with temperature up to 200 K, which could be due to the scattering from the charged impurities[28]. Supplementary data for MR measurements (temperature, bias and back-gate dependence) can be found in Supplementary Note 6.

**Spin diffusion length.** In order to explain our experimental results, we have calculated the MR from the standard theory of spin-injection adapted to lateral devices with tunnelling injectors[20,31,32]. The expected MR is calculated versus the characteristic resistances which are the tunnelling interface resistance ($R_I$) and the channel spin-resistance ($R_N^\star = R_{sq} \cdot l_{sf}/w$) (ref. 31) for a lateral spin-valve in the so-called 'open' geometry (that is, the injected spin may diffuse freely in the channel outwards both contact regions)[32]. The generic formula is given by:

$$\text{MR} = \frac{\Delta R}{R} = \frac{4\gamma^2 R_I^2 R_N^\star / [R_I(1-\gamma^2) + R_N^\star \frac{L}{2l_{sf}}]}{(2R_I + R_N^\star)^2 \exp\left(\frac{L}{l_{sf}}\right) - (R_N^\star)^2 \exp\left(-\frac{L}{l_{sf}}\right)} \quad (2)$$

where $\gamma$ is the tunnel spin polarization (injector and detector), chosen to be 0.5. (The spin polarization $\gamma$ is deduced from the tunnel magnetoresistance (TMR) of a magnetic tunnel junction with amorphous Al$_2$O$_3$ barrier by $\gamma = [\text{TMR}/(2+\text{TMR})]^{1/2} = 0.5$ with TMR = 70%. We assume that the MgO tunnel barrier is amorphous on MoS$_2$). $L$ is the channel length (450 nm) and $l_{sf}$ is the spin diffusion length (SDL), found to be close to 235 nm from refined analyses (detailed hereafter). The channel resistance $R_N$ possesses a certain relationship with its spin-resistance $R_N^\star$ according to $R_N = R_N^\star \cdot L/l_{sf}$, and this makes $R_N$ close to $R_N^\star$ when $L \approx l_{sf}$. As shown in Fig. 5a, one can observe a theoretical maximum value of MR of about 1% in a narrow window when the spin-dependent tunnel resistance $R_I$ is almost equal to the channel spin-resistance $R_N^\star$. This condition corresponds to a perfect balance between the spin-injection rate ($1/R_I$) and the spin relaxation rate ($1/R_N^\star$) giving a same maximum of MR for the whole range of $R_I$ and $R_N^\star$. To better demonstrate the relationship between $R_I$ and $R_N^\star$, we have plotted in Fig. 5b the MR as a function of the ratio $R_I/R_N^\star$ for different $l_{sf}$. The maximum MR increases with $l_{sf}$ due to the evanescent exponential prefactor describing the spin-memory loss scaling with $\exp(-L/l_{sf})$, however always being localized around $R_I/R_N^\star = 1$. From this optimal condition, a larger $R_I$ would reduce the rate of spin-injection compared

to the spin-flip rate and consequently reduce the spin-accumulation in MoS$_2$. On the contrary, a larger $R_N^\star$ gives rise to a spin-backflow process by which the spin would relax in the ferromagnet (Co) and resulting in a reduced injected spin-current and MR from the optimum condition of equality between the tunnel contact resistance and channel spin-resistance (see ref. 32 and Supplementary Note 7).

Assuming the balance condition $R_I = R_N^\star$, the spin diffusion length ($l_{sf}$) estimated from formula (2) for a 1% MR is close to 235 nm. This appears as a lower bound because of the relative high spin-polarization $\gamma$ (0.5) chosen for Co/MgO. Other tunnelling processes than direct tunnelling would give rise to small MR by several orders of magnitude. In particular, sequential tunnelling processes via interface states (IS) between MgO and MoS$_2$, like observed in Co/Al$_2$O$_3$/GaAs structures[33] and responsible for spin-accumulation amplification, would be detrimental for MR. (In a spin injection/transport/detection experiments, the level of spin accumulation generated in MoS$_2$ through spin-injection is generally reduced by the presence of intermediate states[33]. This would result in an overall reduction factor of $(R_{MgO})^2 \cdot R_{MS}^\star/(R_{SC})^3$ even in the case of infinite spin-relaxation time on these IS. Here, $R_{MgO}$ is the MgO barrier resistance (about 30 kΩ), $R_{MS}^\star$ is the spin-resistance of MoS$_2$ ($R_{sq} \cdot l_{sf}/w = 20$–60 kΩ) and $R_{SC}$ is the Schottky resistance (MΩ).) This excludes a sequential tunnelling to form the interface resistance and emphasizes that a direct tunnelling process through one MgO and Schottky composite barrier should be taken into account. Another important conclusion is that an $l_{sf}$ of 235 in multilayer MoS$_2$ at low temperature is already ten times larger than the value predicted in ML MoS$_2$ taking into account the DP spin-depolarization mechanism[14]. To strengthen that point, the inset of Fig. 5b displays the characteristic MR versus channel distance acquired between different electrodes (E1–2, E2–3 and E1–3) together with the simulation curve obtained for $l_{sf} = 235$ nm. It fingerprints the distance-dependence of the spin-injection/detection process beyond other spurious effects. The exponential decay of MR is in good agreement with the simulated MR versus the channel distance. This validates the estimated long spin diffusion length at least larger than 200 nm in our 6ML MoS$_2$. From the extracted mobility of MoS$_2$ channel ($\mu = 6\,\text{cm}^2\,\text{V}^{-1}\,\text{s}^{-1}$) at 12 K ($V_g = +20$ V, $V_{ds} = -1$ V), one can estimate the spin lifetime $\tau_{sf}$ to be 46 ns from $\tau_{sf} = l_{sf}^2/(2D) = l_{sf}^2 e/(2\mu k_B T)$, where $D$ is the diffusion constant. Remarkably, this spin lifetime is one order of magnitude larger than the electron spin relaxation time recently

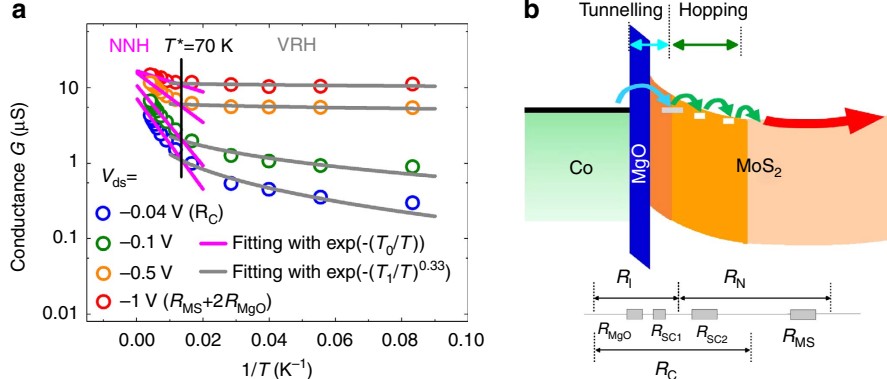

**Figure 6 | Evidence of hopping transport in the contact region.** (**a**) Arrhenius plot of the temperature dependent conductance (symbols) at different $V_{ds}$ from Fig. 4d and the fitting results by different hopping models (grey and pink lines). Two hopping regimes are clearly separated by $T^\star$ (vertical line). (**b**) Band diagram of the Schottky contact region of the MoS$_2$ device. The device can be divided into three regions. The direct tunnelling region consists of the MgO tunnelling barrier ($R_{MgO}$) and one part of Schottky contact ($R_{SC1}$) taken as a whole. The second region is in the tail part of depletion layer where electrons transport in a hopping behaviour ($R_{SC2}$). The third region is the region where electrons either transport in the MoS$_2$ channel by a hopping behaviour or transport in the MoS$_2$ conduction band ($R_{MS}$), depending on the carrier density.

measured in ML MoS$_2$ by optical Kerr spectroscopy[34]. Since the multilayer MoS$_2$ possesses an indirect band gap restricting the ability to observe spin relaxation by optical means, the electric spin-injection/detection method provides an efficient alternate method to probe quantitatively the spin relaxation mechanisms in such compound.

## Discussion

As mentioned above, an optimized MR occurs at the balance condition of a tunnel contact resistance approaching the channel spin-resistance. It seems however impossible to perfectly fulfil such condition in our device at low $V_{ds}$ bias. As shown in Fig. 5a, in the best MR situation, the MoS$_2$ channel resistance ($R_{MS}$) is estimated to be about 150 kΩ. The contact resistance in the investigated range of bias and temperature lies in the MΩ range, well beyond the characteristic threshold, and should then exclude any MR. How may one then reconcile with the standard spin-injection model? To clarify that point, let us focus on the $T$-dependence of the conductance as a fingerprint of the electronic hopping process involved in the transport. Figure 6a displays the Arrhenius plot of the $T$-dependence conductance at different $V_{ds}$. It becomes obvious that the charge transport may be described by two distinct mechanisms in the respective high and low temperature regimes, with a characteristic threshold at $T^\star \sim 70$ K. For $T > T^\star$, the transport is dominated by the nearest-neighbour hopping (NNH) with a conductivity varying like $G \sim \exp(-T_0/T)$. For $T < T^\star$, the conductance can be fitted by a 2D variable-range hopping (VRH) equation according to $G \sim \exp[-(T_1/T)^{0.33}]$. Such characteristic $T$-dependence has been observed in many low-dimensional systems and is a signature of hopping transport via localized states[29,35,36]. In MoS$_2$ system, it is reported that the sulphur vacancies can introduce localized donor states inside the bandgap[29]. The two temperature regions for the different hopping regimes are even more pronounced at small $|V_{ds}|$ (0.04 V) when the contact resistance dominates the total resistance. This highlights a transport dominated by hopping in the contact region more than in the MoS$_2$ channel by itself.

In this scenario, we propose that the contact region may be constituted by three different zones (Fig. 6b): (i) the chemical MgO tunnel barrier, (ii) the strongly depleted zone underneath MgO (SC1) playing the role of an additional composite tunnel barrier and (iii) the tail of the MoS$_2$

depletion zone where the electronic conduction is ensured by hopping mechanism (SC2). When one considers an inhomogeneous spin-channel made of two parts: a depletion part 'D' of length $t$ and a semi-infinite channel part 'B', the effective channel spin-resistance reads $R_N^\star = R_{eff}^\star = (R_B^\star R_D^\star \coth(t/l_{sf}^D) + (R_D^\star)^2)/(R_B^\star + R_D^\star \coth(t/l_{sf}^D))$, where $R_B^\star$ is the 'bulk' spin-resistance of the channel and $R_D^\star = \rho_D l_{sf}^D$ ($l_{sf}^D$ is the spin diffusion length in the depletion region). For thin depletion case ($t \ll l_{sf}^D$), like considered here, $R_D^\star \coth(t/l_{sf}^D) \gg R_B^\star$, leads to $R_{eff}^\star \approx R_B^\star + R_D^\star \tanh(t/l_{sf}^D) \approx R_{MS} + R_{SC2}$.

The observation of MR at low bias goes in favour of an impedance matching at the level of the contact region, by acting with back-gate voltage, however without preventing spin injection and spin transport in the MoS$_2$ channel. The impedance matching is achieved when the ratio between the tunnelling injector resistance $R_I$ ($R_{MgO} + R_{SC1}$) and the channel effective spin-resistance ($R_{eff}^\star$) close to $R_N$ ($R_{SC2} + R_{MS}$) approaches unity from small to large values depending on which part of MoS$_2$ consists in the SC1 or SC2 regions. At low bias, the impedance matching in our Co/MgO/MoS$_2$ devices is then only possible thanks to the high resistivity region (SC2) in the channel. Since the maximum MR is expected at about equal value of $R_I$ and $R_{eff}^\star$, the non-linear variation of the observed MR versus $V_g$ reflects the balance ratio between $R_I$ and $R_{eff}^\star$. When $V_g$ increases from $+8$ V to $+20$ V, due to the shift of $E_F$ in the MoS$_2$ band gap, the faster decay of the MoS$_2$ channel resistance ($R_{MS}$) compared to the tunnel injector resistance ($R_I$) contributes to the enhancement of MR (Fig. 3d). One can also invoke the increase of the electron mobility (Fig. 2f) and spin-flip time. For larger $V_g$, due to the shrinking of the Schottky depletion layer, a continuous decreasing of $R_{SC2}$ cannot fulfil anymore the balance condition between $R_I$ and $R_{eff}^\star$, and results in the drop of MR. When increasing the bias or the temperature, the electrical field[37] or thermal activation energy[38] can favour electron hopping via localized states, especially for the variable-range hopping process and can effectively reduce $R_{SC2}$. This explains the drop of MR with increase of $V_{ds}$ and $T$ is also due to the deviation of the impedance balance condition.

In conclusion, we have demonstrated the electrical spin injection and detection through the conduction band of a 450 nm long, 6ML thick MoS$_2$ channel. From the systematic studies of the bias, temperature and back-gate voltage dependence of MR, it is found that the hopping via localized states in the contact depletion region plays a key role to keep the balance

condition between the interface tunnelling resistance and the channel resistance, which is mandatory for the observation of the two-terminal MR. Moreover, the electron spin-relaxation is found to be greatly suppressed in the multilayer $MoS_2$ channel for an in-plane spin injection geometry. The underestimated long spin diffusion length ($\sim 235$ nm) could open a new avenue for spintronic applications using multilayer TMDs.

## Methods

**Nano-device fabrication.** The $MoS_2$ flakes were exfoliated from a bulk crystal (SPI Supplies), using the conventional micro-mechanical cleavage technique, onto a clean $SiO_2$ (100 nm)/n + + -Si substrate. First e-beam lithography (Raith-150) was performed to define the four electrodes on the selected flake. Then the sample was introduced into a molecular beam epitaxy system to deposit the FM electrodes, which consists of 2 nm MgO, 10 nm Co and 10 nm Au. After deposition and lift-off, a second e-beam lithography procedure was used to define the four large pads for electrical connection. Then Ti(10 nm)/Au(190 nm) was thermally evaporated in a PLASSYS MEB400s system for the large pads. After lift-off, the device was annealed in the vacuum at 200 °C for 1 h followed by the coverage of 10 nm MgO protection layer. To check the thickness of $MoS_2$ flake and the distance of channel, we have performed atomic force microscopy characterization in tapping mode on the device. In order to precisely extract the flake thickness, Gaussian fitting of the distribution of height has been employed.

**Magneto-transport measurements.** The magneto-transport measurements have been performed in a cryostat varying temperature from 12 to 300 K with a maximum magnetic field of 4kOe. For the device presented in the main text, in order to reach a well-defined AP magnetic configuration, a magnetic field was applied at a 45° angle to the electrodes. Magnetic domains are then generated through the reservoir of the large triangle areas of the electrodes (Fig. 1a) before propagating towards the injector and detector regions above the $MoS_2$ flake[39] (see more micromagnetic simulations in Supplementary Note 4). For the back-gated two-terminal spin-valve measurement as described in Fig. 1d, we have used a Keithley 2400 to apply the drain-source bias $V_{ds}$, and used a Keithley 6487 picoamperometer to measure the drain-source current $I_{ds}$. At the same time, another Keithley 2400 was employed to apply the back-gate voltage $V_g$.

**Data availability.** The data that support the findings of this study are available from the corresponding author on request.

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

## Acknowledgements

We thank Prof Mingwei Wu for helpful discussions in the spin relaxation in $MoS_2$. We also acknowledge François Montaigne, Daniel Lacour and Michel Hehn for discussion of manuscript. This work is supported by French National Research Agency (ANR) MoS2ValleyControl project (Grant No. ANR-14-CE26-0017-04), ANR Labcom project (LSTNM) and the joint ANR-National Natural Science Foundation of China (NNSFC) ENSEMBLE project (Grants No. ANR-14-CE26-0028-01 and No. NNSFC 61411136001). X.M. acknowledges Institut Universitaire de France and P.R. thanks the grant NEXT No ANR-10- LABX-0037 in the framework of the Programme des Investissements d'Avenir. Experiments were performed using equipment from the platform TUBE-Davm funded by FEDER (EU), ANR, the Region Lorraine and Grand Nancy.

## Author contributions

Y.L. coordinated the research project and designed sample structures. S.L., H.Y., G.W. and Y.L. fabricated $MoS_2$ samples. Y.L., S.M.-M., S.L., H.Y. and B.T. contributed to develop e-beam lithography for the device fabrication. S.L. and Y.L. performed magneto-transport measurements. H.J. developed spin diffusion theory. P.L. carried out micromagnetic simulations. Y.L. and H.J. wrote the manuscript, with help of S.L., P.R., X.M., J.-M.G, S.M. and S.P.-W. All authors analysed the data, discussed the results and commented on the manuscript.

## Additional information

**Competing interests:** The authors declare no competing financial interests.

