## [Peer Review File · Nature Communications]

Reviewers' Comments:

Reviewer #1 (Remarks to the Author)

Manuscript presents observation of anomalously long spin relaxation and diffusion length in multilayer TMD material structure. I find it quite an important observation because parameters are found to exceed values of some of the most popular spintronic materials.

So I recommend publication as Nature Comm. after authors consider following optional changes:

1) English needs improvements, e.g. "in-plan" in abstract or sentence in line 41 sounds semantically strange, e.g., "precession along the field" or why precession is due to Dyakonov Perel relaxation? Usually the opposite is considered true.

2) Authors must stress that multilayer restricts ability to observe spin relaxation optically.

3) line 114: I cannot subscribe for this. Spin injection has been studied by many groups before. I suggest to drop this sentence or at least the word "first" to avoid possible conflict.

Reviewer #2 (Remarks to the Author)

In this work the authors report on their study of a fully electronic spin injection and detection in a MoS₂ lateral spin valve (LSV). The integration of di-chalcogenides with spintronics is an ongoing challenge with much interesting physics and applications. To the best of my knowledge, a full electronic injection and detection has not been reported yet for this family, thus the subject of this work is important and timely.

The authors' main claim is that they were able to observe the magnetoresistance from LSV of a multi-layer MoS₂ device, and extract an estimation for the spin diffusion length, for which they give a lower bound. I find the data provided to be sufficient for the support of the claim. However there are critical problems with the paper that must be addressed and clarified before being able to fully assess the paper, and before it is suitable for publication.

First, I want to stress that the paper is very difficult to read and follow. The authors jump back and forth between figures, constantly talking about different parts of different figures, many times not in the chronological order of appearance. This made the reading very difficult from the eyes of a first-time reader this was overwhelming. The authors must make an effort to reconstruct the paper in a reader-friendly way.

This made assessing of the manuscript more difficult.

As pointed out through the comments below, there is some missing information that was needed to be able to verify the accuracy of the claims.

I would recommend to separate in the manuscript the proof for the main claims regarding full electronic measurement of spin injection in MoS₂ and SDL attained by impedance matching, to the more technical and some-times speculative explanations of the V_{ds} and V_g dependence of the signal.

Additional specific comments and questions appear following.

81: For some reason the authors regard the term of the MgO as a constant contact resistance, which is misleading, since tunneling current is of course bias dependent.

85: The plot of Fig 2e is clearly not linear with channel distance. This is since the plot is for constant V_{ds} . For a constant channel length and since the tunnel/SB changes exponential with voltage, increasing above some voltage will result in a very small change of voltage on the contact, and most of it falling on the channel, leading to a (nearly) linear IV behavior. It is wrong

to analyze this system assuming a constant contact resistance as shown in the inset of 2f. I suggest the authors consider doing this analysis for a constant current, I_{ds} . The rationale is that for tunneling or SB junction the voltage drop is a function of current. In this way they can ensure that there is the same voltage drop on the contacts, and the change in resistance as a function of distance will be linear, enabling to extract the channel resistance. This will also make the MR analysis with different channel length more relevant, especially since all these features change with V_{ds} .

The main part of fig 2f is to find the best V_{ds} for impedance matching, so showing where the channel resistance is equal to the contact resistance is helpful.

At +10V from Fig 3C it is clear that $V_g > V_f$. So it means that one Co contact includes both MgO contact and the effect of the SB while the other has only the Contact. So the contribution is not similar and is not $2R_{MgO}$.

128 and fig 3c: This analysis would provide the SB height only when thermal activation is dominant, i.e. below $\sim 2.6V$. Above this value, the analysis can still be performed, but it does not have the interpretation of barrier height, since tunneling is dominant. I think it would benefit readers if this point is clearly stated.

140: The discussion on change of resistance in MS and contact vs. gate is not clear to me. The contact resistance is voltage dependent and not constant, while the MS resistance should not depend on V_{ds} , only on V_g (assuming far from the pinch-off region). So at any gate one can dial-in the contact resistance by changing V_{ds} , and this should not be a problem.

145: In this part the authors discuss the issue of impedance mismatch problem. Reading this part, I expected that the MR would vanish for low enough V_{ds} , but it remained high. I tried to understand for a long time why the authors disregard this issue. Only upon further reading in the discussion the issue was addressed. Maybe have the entire discussion in the discussion (This is an example to my first comment regarding the readability of the paper).

200: I think the explanation is plausible but highly speculative. I would estimate that VRH would lead to faster spin dephasing times. There are many inconsistency of adding these regions with the rest of the paper.

The small V_{ds} (0.04) should be mainly property of tunneling barrier, and not VRH part.

These additional parts are also voltage dependent, so they undermine all previous analysis of the SB height etc. which disregard their contribution.

I request the authors to add in the supplementary the MR plots of the 2 additional channels they measured. Also please provide at least for refereeing (not needed to add to supplementary) measurements of another device where the MR measurements were reproduced, also include MR of different channel lengths.

FM properties:

Fig s7 of the estimated electrode switching is so different than the results presented throughout the paper, that I think they are not relevant. It would have been better if the authors could provide a simpler 2 probe measurement of the FM electrodes as is often shown in LSV paper to prove the point. The appearance of switching of the first electrode so close to zero field is suspicious, at the least. It is also very peculiar that upon changing the temperature between 23K and 12K the coercive field of one of the electrodes increased from 0.5T to 1T, while the other electrode shows about zero. I think the authors must provide evidence for the magnetic switching behavior of their FM electrodes.

In the sup. the authors explain why they concentrate on the negative V_{ds} region. This does not make much sense. Throughout the paper they discuss that for MR the electrode properties are symmetric. The resistance of I in the model is considered the same, and this would cause much difficulty to attain good matching with the electrodes if each contact is different. Please clarify.

Why do fig 2d,2e re-appear in s1a,b? Additionally, please supply for refereeing graphs similar to s1c,d , but for the $V_g=0$ data. Thank you.

Reviewer #3 (Remarks to the Author)

The authors present an experimental study of the lateral spin transport in multilayer MoS₂, where the electrons are injected from Co/MgO layers. The study is technically sound and interesting. One of the main claims that "This is the first demonstration of electron spin transport signal through the semiconducting MoS₂ in a lateral device" is not fully correct, see for example Nano Lett. 2016, 16, 5792-5797 (Ref. 16). While in that case the spins are injected into WSe₂, it clearly is the analogous case to spin-injection into MoS₂. The measured magnetoresistance effect is however novel enough to warrant publication in Nature Communications.

The following points need to be addressed:

- 1) The authors should provide an explanation for the increase of the mobility with temperature in Fig. 4e.
- 2) A detailed derivation of Eq. 2 should be given in the supporting information
- 3) Can the authors propose possible ways to enhance the rather low magnetoresistance of about 1%? Is there a fundamental device limitation for such a low magnetoresistance?

We sincerely thank all referees for their careful reading of our manuscript and very insightful and constructive comments. Taking into account of all their comments, we have addressed all questions and made corresponding modifications in the new manuscript. We hope the below responses can satisfy the referees.

Reviewer #1 (Remarks to the Author):

1) *English needs improvements, e.g. "in-plan" in abstract or sentence in line 41 sounds semantically strange, e.g., "precession along the field" or why precession is due to Dyakonov Perel relaxation? Usually the opposite is considered true.*

Answer: We have improved English writing according to the suggestion of Referee. In Page 2, Line 14, the structure of sentence is improved for better understanding the precession induced by the effective magnetic field due to the DP spin relaxation mechanism.

2) *Authors must stress that multilayer restricts ability to observe spin relaxation optically.*

Answer: We have added one sentence in Page 8, Line 7 "Since the multilayer MoS₂ possesses an indirect band gap which restricts the ability to observe spin relaxation optically, the electric spin-injection and detection provides an effective way to probe the spin relaxation mechanism in the multilayer MoS₂ channel."

3) *line 114: I cannot subscribe for this. Spin injection has been studied by many groups before. I suggest to drop this sentence or at least the word "first" to avoid possible conflict.*

Answer: We remove the word "first" in the sentence.

Reviewer #2 (Remarks to the Author):

First, I want to stress that the paper is very difficult to read and follow. The authors jump back and forth between figures, constantly talking about different parts of different figures, many times not in the chronological order of appearance. This made the reading very difficult from the eyes of a first-time reader this was overwhelming. The authors must make an effort to reconstruct the paper in a reader-friendly way. This made assessing of the manuscript more difficult.

Answer: We have changed the sequence of figures in Fig.2, and separated the Fig.4 into two figures (add Fig.6). Now the figures appear in a chronological order for easy reading. Moreover, we have reconstructed the paper with specifying some titles at the beginning of each sections and subsections for a better clarity.

I would recommend to separate in the manuscript the proof for the main claims regarding full electronic measurement of spin injection in MoS₂ and SDL attained by impedance matching, to the more technical and some-times speculative explanations of the V_ds and V_g dependence of the signal.

Answer: The structure of paper has been carefully considered and been constructed as following:

1. In the first part, we present the transport characterization on MoS₂ channel vs. Tunnel contacts;
2. In the second part, we present the core results of this paper: all magnetoresistance measurement results with V_g , V_{ds} and temperature dependence of MR;
3. In the third part, we performed theoretic simulation and established models to estimate the spin diffusion length in MoS₂ by electrical (MR) methods;
4. In the fourth part, we discuss on how to understand the experimental MR results with the theoretic simulation in the optimal impedance mismatch condition.

81: For some reason the authors regard the term of the MgO as a constant contact resistance, which is misleading, since tunneling current is of course bias dependent.

Answer: The resistance of device is divided into several parts. At low V_{ds} ($|V_{ds}| < 0.15V$), the MgO tunneling resistance is only very small part compared to Schottky resistance in the contact region. To simplify, we consider it as a constant. In the revised manuscript, we have removed the figure to extract R_{MgO} . However, note that the experiments demonstrating MR have been done at sufficiently low bias (-0.04V) in order to reject any possible spin-depolarization mechanisms induced by hot electron processes. In that bias range, the MgO resistance can be considered as a constant.

85: The plot of Fig 2e is clearly not linear with channel distance. This is since the plot is for constant V_{ds} . For a constant channel length and since the tunnel/SB changes exponential with voltage, increasing above some voltage will result in a very small change of voltage on the contact, and most of it falling on the channel, leading to a (nearly) linear IV behavior. It is wrong to analyze this system assuming a constant contact resistance as shown in the inset of 2f.

I suggest the authors consider doing this analysis for a constant current, I_{ds} . The rationale is that for tunneling or SB junction the voltage drop is a function of current. In this way they can ensure that there is the same voltage drop on the contacts, and the change in resistance as a function of distance will be linear, enabling to extract the channel resistance. This will also make the MR analysis with different channel length more relevant, especially since all these features change with V_{ds} .

Answer: We thank the referee for this very insightful comment. Indeed, the variation of resistance vs. channel distance/width at different V_{ds} is not exactly linear. This is because there is one part of the MoS₂ depletion region contributing to the contact resistance. This resistance can be varied depending on the partial voltage drop on it. With the increase of channel length, the voltage drop in the MoS₂ depletion region changes, which introduces the non-linear behavior. By using I_{ds} instead of V_{ds} , the linear behavior is much improved.

In this revised version of the manuscript, we have used identical I_{ds} method to extract R_C and R_{MS} , which is shown in Fig. 2c and 2d in main text. We also rewrite the part S1 in supplementary information to compare the two methods. The inset of 2f in last version has been removed.

At +10V from Fig 3C it is clear that $V_g > V_f$. So it means that one Co contact includes both MgO contact and the effect of the SB while the other has only the Contact. So the contribution is not similar and is not $2R_{MgO}$.

Answer: We agree that the contact resistance which includes mainly two MgO barriers and one Schottky depletion zone close to the injector part. At large V_{ds} , the width of depletion zone is greatly suppressed which effectively reduces the contact resistance. We assume that the contact resistance approaches $2R_{MgO}$ with large V_{ds} and large V_g .

In the revised manuscript, we have specified on Fig. 3d, that the measured resistance R consists in $R_{MS}+R_C$ instead of $R_{MS}+2R_{MgO}$ when $V_{ds}=-1V$. In contrast, at $V_g=+20V$, (Fig. 4d) one can simplify into $R_{MS}+2R_{MgO}$, and the temperature dependence mainly reveals the change in R_{MS} .

128 and fig 3c: This analysis would provide the SB height only when thermal activation is dominant, i.e. below $\sim 2.6V$. Above this value, the analysis can still be performed, but it does not have the interpretation of barrier height, since tunneling is dominant. I think it would benefit readers if this point is clearly stated.

Answer: We thank the referee for this comment. We have added this explanation into the manuscript Page 5, Line 27.

140: The discussion on change of resistance in MS and contact vs. gate is not clear to me. The contact resistance is voltage dependent and not constant, while the MS resistance should not depend on V_{ds} , only on V_g (assuming far from the pinch-off region). So at any gate one can dial-in the contact resistance by changing V_{ds} , and this should not be a problem.

Answer: We have performed measurements to extract R_C and R_{MS} with only two back-gate voltages: 0V and 10V. We found that, the two cases have the same tendency that R_C rapidly decreases with $|V_{ds}|$, while R_{MS} only changes slowly with V_{ds} . Therefore, we can use R_{total} at different V_{ds} to deduce the variation of R_C and R_{MS} . At low V_{ds} , the total resistance R_{total} approaches to R_C , while at high V_{ds} , there is an important contribution of R_{MS} in the total resistance R_{total} .

145: In this part the authors discuss the issue of impedance mismatch problem. Reading this part, I expected that the MR would vanish for low enough V_{ds} , but it remained high. I tried to understand for a long time why the authors disregard this issue. Only upon further reading in the discussion the issue was addressed. Maybe have the entire discussion in the discussion (This is an example to my first comment regarding the readability of the paper).

Answer: In the previous question, we have explained how we construct the structure of manuscript. In the second part, we only present all magnetoresistance measurement results with V_g , V_{ds} and temperature dependence of MR and R without entering into detailed explanation. Then we performed theoretic simulation to estimate spin diffusion length and try to understand the measured MR. Finally, we discussed how to reconcile between experiments and simulations and explain all tendency of MR with V_g , V_{ds} and T .

In the revised manuscript, we explain that the MR is maximum at low bias where the hot electrons spin-depolarization process can be almost suppressed. In that situation, the MR appears on acting on the gate voltage when the tunneling contact equals the contact spin-resistance close to the contact resistance. There is a particular gate voltage (+20V) for which the condition is fulfilled. This condition of a high contact resistance is possible due to the depletion process

underneath the tunneling injector introducing location of hopping (increasing the resistance) without preventing the injection in the bulk MoS₂ (more favorable for spin transport).

200: I think the explanation is plausible but highly speculative. I would estimate that VRH would lead to faster spin dephasing times. There are many inconsistency of adding these regions with the rest of the paper.

Answer: What we would like to show is the evidence of hopping in the contact region. From the analysis of R^{-1} vs. T^{-1} at different V_{ds} , we found that the VRH process is quite pronounced at small V_{ds} , where the contact resistance is dominant. In fact, the standard spin diffusion model does not taken account of the spin dephasing via VRH. The VRH might lead to faster spin dephasing, however we still believe it is not a major issue to determine the spin injection/detection efficiency in the actual two-terminal device. In the future, if we can even improve the channel mobility, the depletion zone with VRH process will not be necessary to keep the impedance balance condition, the MR can be even enhanced due to the suppression spin-dephasing in the VRH region.

The small V_{ds} (0.04) should be mainly property of tunneling barrier, and not VRH part.

These additional parts are also voltage dependent, so they undermine all previous analysis of the SB height etc. which disregard their contribution.

Answer: For the analysis of SB height, all measurements are performed between 180K-240K with $|V_{ds}|$ (0.4-1V), where thermionic emission takes place (see SI part S5). However, VRH and MR disappear when T is larger than 70K. The evidence of VRH with small V_{ds} (0.04V) below 70K is to prove that there exists a hopping zone in the contact region, and this does not influence on the analysis of SB height.

I request the authors to add in the supplementary the MR plots of the 2 additional channels they measured. Also please provide at least for refereeing (not needed to add to supplementary) measurements of another device where the MR measurements were reproduced, also include MR of different channel lengths.

Answer:

1. Two different channel MR curves

As requested by the referee, we have added two supplementary data set corresponding to different channel lengths showing up the characteristic MR exponential decrease expected from spin-valve effects. This clearly indicates that the signal does not origin from other spurious effects (channel magnetoresistance, anisotropy of magnetoresistance of ferromagnetic electrode, tunneling anisotropic magnetoresistance). The exponential decay of MR is in good agreement with the simulated MR vs. the channel distance. This validates the estimated long spin diffusion length at least larger than 200nm in our 6ML MoS₂.

Figure R1: (a-c) I_{ds} - V_{ds} characteristics measured between different electrodes at 12K with applying different back-gate voltage: (a) E1-E2 electrodes with channel length of 450nm; (b) E2-E3 electrodes with channel length of 650nm; (c) E1-E3 electrodes with channel length of 1400nm. (d-e) Magneto-resistance response of the device with $V_{ds}=-0.1V$ and $V_g=+20V$ at 12K for (d) E1-E2, (e) E2-E3 and (f) E1-E3 electrodes.

2. Magneto-resistance of MoS2 device-2

For the device presented in the main text, the four electrodes on MoS2 have almost identical widths. To improve the magnetic properties for antiparallel configuration, we have designed a second device with electrodes with different widths, as shown in Fig. R2a. In addition, the area of the ferromagnetic contact is sufficiently small to ensure the mono-domain magnetic structure. The MoS₂ channel distance between electrodes E1 and E2 is measured as 670nm, and the thickness of the flake is also about 4.3nm (6MLs). Fig. R2b shows the characterization of I_{ds} as a function of V_{ds} with different V_g , and Fig. R2c shows the variation of I_{ds} as a function of V_g with different V_{ds} . The features of these curves are almost identical to the device presented in the main text. Finally, we have measured the magneto-resistance response curve as shown in Fig. R2d. About 0.88% of MR has been obtained with $V_{ds}=0.1V$ and $V_g=+20V$ at 12K. This means that the observation of MR is not specific to only one device. The growth and e-beam lithography procedures yield reproducible results. For this device, unfortunately we do not check MR with different channel distance.

Figure R2: (a) Optical microscopy image of the device-2. The region with ferromagnetic (FM) contact consists of MgO(2nm)/Co(10nm)/Au(10nm) and the region with nonmagnetic (NM) contact consists of Ti(10nm)/Au(190nm). (b) I_{ds} - V_{ds} characteristics between E1 and E2 measured at 12K with applying different V_g . (c) I_{ds} - V_g characteristics between E1 and E2, measured at 12K with applying different V_{ds} . (d) Magneto-resistance response of the device $V_{ds}=0.1V$, $V_g=+20V$ at 12K.

FM properties:

Fig s7 of the estimated electrode switching is so different than the results presented throughout the paper, that I think they are not relevant. It would have been better if the authors could provide a simpler 2 probe measurement of the FM electrodes as is often shown in LSV paper to prove the point. The appearance of switching of the first electrode so close to zero field is suspicious, at the least. It is also very peculiar that upon changing the temperature between 23K and 12K the coercive field of one of the electrodes increased from 0.5T to 1T, while the other electrode shows about zero. I think the authors must provide evidence for the magnetic switching behavior of their FM electrodes.

Answer: We thank the referee for that questions and we need to clarify that point as explained below. If our understanding is correct, this requires that each FM contact has two electrical connection so that one can probe the magnetic state by AMR effect. However, in our device, each FM electrode only possesses a single electrical contact preventing a 2 probe measurement. All our simulation with Mumax3 was performed under 0K. The simulations show a 100Oe large AP plateau which can be reached with the geometry of our electrodes. This is in agreement with our experiment results (200-500 Oe). With the increase of temperature from 12K to 40K, the AP plateau width decreases from 500 to 200 Oe. This should be related to the decrease of the coercivity of the electrode from 700 to 400 Oe. The temperature effect on domain propagation and coercivity has already extensively studied [Chin. Phys. B 24, 034501 (2015), PRB 63, 104418

(2001), Nat.Mat. DOI:10.1038/NMAT4248]. The thermal activation energy can effectively de-pinning the propagation of domain, thus decreasing the coercivity, this agrees well with our finding that AP plateau width decreases when T increasing, but however not in the scale between 0.5T and 1T like mentioned by the referee.

In the sup. the authors explain why they concentrate on the negative V_{ds} region. This does not make much sense. Throughout the paper they discuss that for MR the electrode properties are symmetric. The resistance of I in the model is considered the same, and this would cause much difficulty to attain good matching with the electrodes if each contact is different. Please clarify If the two contact is different, how to attain good matching ? The matching is between the interface tunneling resistance and channel resistance. If the two interface tunneling resistance is not the same, how to derive a balance to get good match?

Answer: Experimental measurements show that the asymmetry in the I_{ds} - V_{ds} curves are not so pronounced. The asymmetry may only originate from the triangle shape of the MoS_2 flake, so that the contact area is slightly different. However, the tunneling contacts, even not exactly identical, display about the same Schottky profile and potential drop within the so-called contact region. From the point of view of MR, as explained in the supplementary material (S7), unidentical contact sizes lead to different spin-dependent tunneling resistance and lead to a (slight) decrease of MR vs. the asymmetry. However the decrease of asymmetry is relatively small, of the order of 10% of decrease of MR for a factor of 2 in the asymmetry of the tunneling resistance and of the order of 50% in MR for a factor of 5 in the asymmetry of the tunneling resistance.

Why do fig 2d,2e re-appear in s1a,b? Additionally, please supply for refereeing graphs similar to s1c,d , but for the $V_g=0$ data. Thank you.

Answer: We have added the $V_g=0$ data in the supplementary information (S1). We have found the the linearity of R vs. channel distance is even worth than that of $V_g=+10V$ if we use identical V_{ds} method. Therefore, we present here only the extraction of R_C and R_{MS} with identical I_{ds} method. The results show a similar behavior as that of $V_g=+10V$. R_C decreases rapidly with increase of $|V_{ds}|$ and it dominates the total resistance at low $|V_{ds}|$. R_{MS} also decreases with increasing $|V_{ds}|$. The contribution of R_{MS} in R_{total} increases with $|V_{ds}|$ and saturates at 37% when $|V_{ds}|$ is larger than 0.25V.

Reviewer #3 (Remarks to the Author):

The authors present an experimental study of the lateral spin transport in multilayer MoS_2 , where the electrons are injected from Co/MgO layers. The study is technically sound and interesting. One of the main claims that "This is the first demonstration of electron spin transport signal through the semiconducting MoS_2 in a lateral device" is not fully correct, see for example Nano Lett. 2016, 16, 5792-5797 (Ref. 16). While in that case the spins are injected into WSe_2 , it clearly is the analogous case to spin-injection into MoS_2 . The measured magnetoresistance effect is however novel enough to warrant publication in Nature Communications.

Answer: We remove the word "first" in the sentence.

The following points need to be addressed:

- 1) The authors should provide an explanation for the increase of the mobility with temperature in Fig. 4e.

Answer: The behavior of the temperature dependent mobility in our work is similar to that of published paper [Nat. Mat. 12, 815 (2012)], where the temperature dependence is characterized by a peak at about 200K. Below 200K, the mobility increases with the increase of temperature, which can be explained by the mobility limitation by scattering from the charged impurities [Sze, S. M. & Ng, K. K. Physics of Semiconductor Devices (Wiley, 2007)]. Above 200K, the decrease of mobility with temperature is due to the electron-phonon scattering at higher temperature [PRB 85, 115317 (2012)].

- 2) A detailed derivation of Eq. 2 should be given in the supporting information

Answer: We have added a detailed description on the derivation of spin-injection theory in supplementary information (S7) with development beyond the actual state-of-the art in an international context.

- 3) Can the authors propose possible ways to enhance the rather low magnetoresistance of about 1%? Is there a fundamental device limitation for such a low magnetoresistance?

Answer: The possible ways to enhance the MR are:

1. Improving the contact quality, reduce contact resistance, reduce the defect induced spin dephasing;
2. Increasing mobility of channel, reduce the channel resistance, reduce hopping region spin dephasing;
3. Increasing the spin-injection polarization by using other FM materials as injector;
4. Reducing SOC effect due to asymmetry of interface, reducing the substrate effect.
5. Patterning smaller devices with closer tunneling spin-injector contacts and smaller channel length.

For future application, the MR should be at least 10-100% at room temperature.

Reviewers' Comments:

Reviewer #2 (Remarks to the Author)

I read the revised manuscript and the authors comments and replies.
I found sufficient answers in most of their replies, and modifications.

There is still a problem with the way the SB transport interpretation section is presented. In part of the manuscript and in the response the authors claim that at low V_{ds} ($<0.15V$) the SB is an important part in the transport. But the SB is a few meV, so I would assume that at some V_{gate} (lets say $>10V$) the SB is not relevant and there are only 2 MgO tunnel barriers. Then the rest of the paper makes sense.

But around lines 94-101 the opposite claim is stated. For $V_g=+10V$ the authors claim that the SB profile is the leading contribution to the contact resistance. In addition, they continue to claim that the tunneling resistance is only weakly dependent on V_{ds} . Of course at some voltage V_{ds} , the change in current through the contact is going to be much more sensitive to V_{ds} than the linear dependence of the MS channel, and then it will seem constant. This should be when the resistances are similar in magnitude (as the authors see), and V_{sd} should be on the order of the MgO barrier height (then the current starts changing fast), which is likely the case for 1V. There is no meaning to the $R_{MgO} \sim 32K\Omega$. It will change with MS channel resistance, and it does not supply any insight to the device properties, probably the contrary. I emphasize that the rest of the paper (e.g. around line 136) the analysis makes sense, and this claim is reversed. I urge the authors to rethink and rewrite this section.

Line 158 should be when $T > 60K$ (there is $<$)?

The result showing the large effect of V_{ds} on the MR is very surprising and interesting. The explanation provided by the authors is rather speculative, as I stated before. But lacking a better alternative, I do not have a problem with it, and the evidence supporting it. Did the authors consider local heating of the MS at low temperatures? This may fit with the MR disappearing at 60K.

Reviewer #3 (Remarks to the Author)

The authors have addressed all points raised in the previous round of review, the manuscript can now be published.

We sincerely thank Referee 2 for his/her comments which help us to improve the quality of paper. We have addressed all his/her questions and made corresponding modifications in the new manuscript. We hope the below responses can satisfy the referee.

Reviewer #2:

There is still a problem with the way the SB transport interpretation section is presented. In part of the manuscript and in the response the authors claim that at low V_{ds} ($<0.15V$) the SB is an important part in the transport. But the SB is a few meV, so I would assume that at some V_{gate} (lets say $>10V$) the SB is not relevant and there are only 2 MgO tunnel barriers. Then the rest of the paper makes sense.

But around lines 94-101 the opposite claim is stated. For $V_g=+10V$ the authors claim that the SB profile is the leading contribution to the contact resistance. In addition, they continue to claim that the tunneling resistance is only weakly dependent on V_{ds} . Of course at some voltage V_{ds} , the change in current through the contact is going to be much more sensitive to V_{ds} than the linear dependence of the MS channel, and then it will seem constant. This should be when the resistances are similar in magnitude (as the authors see), and V_{ds} should be on the order of the MgO barrier height (then the current starts changing fast), which is likely the case for 1V.

There is no meaning to the $R_{MgO} \approx 32K\Omega$. It will change with MS channel resistance, and it does not supply any insight to the device properties, probably the contrary.

I emphasize that the rest of the paper (e.g. around line 136) the analysis makes sense, and this claim is reversed.

I urge the authors to rethink and rewrite this section.

Answer : As we have made a detailed description in SI part S5 for the measurement of the Schottky barrier height (SBH), all the SBH presented in Fig. 3c is extracted from the I_{ds} - V_{ds} characteristics with different V_g from **180K to 240K**. Especially, the SBH is extrapolated from the slope $S(V_{ds})$ values between **0.4-1.0V** at zero V_{ds} . Therefore, the SBH values in Fig. 3c only represent the characteristics of Schottky barrier at high temperature and at large V_{ds} . However, around lines 94-101, we talk about the SB profile is the leading contribution to the contact resistance at low V_{ds} ($<0.15V$) and also at low temperature (12K). In Fig. 2d, it clearly shows that the contact resistance quickly decreases and it becomes comparable to the channel resistance when $|V_{ds}|$ is larger than 0.4V. Therefore, there is no contradiction in our claims. **To avoid the confusing understanding, we have added one sentence in Page 5, Line 23: It is noted that all Φ_b values are extracted from the measurements between 180K to 240K and with V_{ds} from -0.4V to -1V.**

Since the "contact" resistance is made of a pure tunneling part by MgO, a tunnel Schottky part and a MoS₂ depletion part. At $|V_{ds}|=1V$ and $V_g=+20V$ when R_c is almost stable, it approaches $2R_{MgO}$ corresponding to the situation free of depletion zone. This allows us to estimate the MgO resistance. We agree with the referee that MgO resistance varies with V_{ds} . However, in the present work, even at $|V_{ds}|=1V$, the voltage drop on one MgO barrier is not larger than 0.4V (taking account of channel resistance in series). Our previous study of Fe/MgO/Fe magnetic tunnel junctions shows that the conductivity of the junction does not change 5% when the bias is smaller than 0.4V [Y. Lu *et al*, Phy. Rev. B 86, 184420 (2012), Fig.3a].

Line 158 should be when $T > 60K$ (there is $<$)?

Answer : We have corrected this error.

The result showing the large effect of V_{ds} on the MR is very surprising and interesting. The explanation provided by the authors is rather speculative, as I stated before. But lacking a better alternative, I do not have a problem with it, and the evidence supporting it. Did the authors consider local heating of the MS at low temperatures? This may fit with the MR disappearing at 60K.

Answer : Since the current passed through the MoS₂ channel is only in the range of tens of nA, the heating effect can be negligible. In addition, there is no shift of background in the recorded MR curves, which can also eliminate the possibility of reducing MR due to the heating effect.

Reviewers' Comments:

Reviewer #2 (Remarks to the Author)

The authors have answered my questions satisfactorily